# A general-purpose machine-learning force field for bulk and nanostructured phosphorus

Volker L. Deringer [1✉], Miguel A. Caro [2,3] & Gábor Csányi [4]

Elemental phosphorus is attracting growing interest across fundamental and applied fields of research. However, atomistic simulations of phosphorus have remained an outstanding challenge. Here, we show that a universally applicable force field for phosphorus can be created by machine learning (ML) from a suitably chosen ensemble of quantum-mechanical results. Our model is fitted to density-functional theory plus many-body dispersion (DFT + MBD) data; its accuracy is demonstrated for the exfoliation of black and violet phosphorus (yielding monolayers of "phosphorene" and "hittorfene"); its transferability is shown for the transition between the molecular and network liquid phases. An application to a phosphorene nanoribbon on an experimentally relevant length scale exemplifies the power of accurate and flexible ML-driven force fields for next-generation materials modelling. The methodology promises new insights into phosphorus as well as other structurally complex, e.g., layered solids that are relevant in diverse areas of chemistry, physics, and materials science.

[1] Department of Chemistry, Inorganic Chemistry Laboratory, University of Oxford, Oxford OX1 3QR, UK. [2] Department of Electrical Engineering and Automation, Aalto University, Espoo 02150, Finland. [3] Department of Applied Physics, Aalto University, Espoo 02150, Finland. [4] Engineering Laboratory, University of Cambridge, Cambridge CB2 1PZ, UK. ✉email: volker.deringer@chem.ox.ac.uk

The ongoing interest in phosphorus[1] is partly due to its highly diverse allotropic structures. White P, known since alchemical times, is formed of weakly bound $P_4$ molecules[2], red P is an amorphous covalent network[3–5] and black P can be exfoliated to form monolayers, referred to as phosphorene[6,7], which have promise for technological applications[8]. Other allotropes include Hittorf's violet and Ruck's fibrous forms, consisting of cage-like motifs that are covalently linked in different ways[9–11], P nanorods and nanowires[12–14] and a range of thus far hypothetical allotropes[15–18]. Finally, liquid P has been of fundamental interest due to the observation of a first-order transition between low- and high-density phases[19–21].

Computer simulations based on quantum-mechanical methods have been playing a central role in understanding P allotropes. Early gas-phase computations were done for a variety of cage-like units[22] and for simplified models of red P[23]; periodic density-functional theory (DFT) with dispersion corrections served to study the bulk allotropes[24–27]. DFT modelling of phosphorene quantified strain response[28], defect behaviour[29] and thermal transport[30]. Higher-level quantum-chemical investigations were reported for the exfoliation energy of black P[31,32], and the latter will be a central theme in the present study as well. For the liquid phases, DFT-driven molecular dynamics (MD) were done in small model systems with 64–128 atoms per cell[33–36].

Whilst having provided valuable insight, these prior studies have been unavoidably limited by the computational cost of DFT. Empirically fitted force fields (interatomic potential models) require much fewer computational resources and have therefore been employed for P as well. Recently, different approaches have been used to parameterise force fields specifically for phosphorene[37–40]. For example, a ReaxFF model was used to study the exfoliation of black P, notably including the interaction with molecules in the liquid phase[41]. However, these empirically fitted force fields can only describe narrow regions of the large space of atomic configurations, which poses a major challenge when very diverse structural environments are present: for example, force fields developed specifically for black P or phosphorene would not be expected to properly describe the liquid phase(s).

Machine-learning (ML) force fields are an emerging answer to this problem[42–48], and they are increasingly used to solve challenging research questions[49–51]. The central idea is to carry out a number of reference computations (typically, a few thousand) for small structures, currently normally based on DFT, and to make an ML-based, non-parametric fit to the resulting data. Alongside the choice of structural representation and the regression task itself, a major challenge in the development of ML force fields is that of constructing a suitable reference database, which must cover relevant atomistic configurations whilst having sufficiently few entries to keep the data generation tractable. Although key properties (such as equations of state and phonons) of crystalline phases can now be reliably predicted with these methods[52], and purpose-specific force fields can be fitted on the fly[53], it is still much more challenging to develop general-purpose ML force fields that are applicable to diverse situations out-of-the-box—to a large extent, this is enabled (or precluded) by the reference data. Indeed, when fitted to a properly chosen, comprehensive database, ML force fields can describe a wide range of material properties with high fidelity[49,50], while being flexible enough for exploration tasks, such as structure prediction[54–57]. Phosphorus has been an important demonstration in the latter field more recently, when we constructed a Gaussian approximation potential (GAP) model through iterative random structure searching (RSS) and fitting[58].

In the present work, we introduce a general-purpose GAP ML force field for elemental P that can describe the broad range of relevant bulk and nanostructured allotropes. We show how a general reference database can be constructed by starting from an existing GAP–RSS model and complementing it with suitably chosen 3D and 2D structures, thus combining two database-generation approaches that have so far been largely disjoint, and giving exquisite (few meV per atom) accuracy in the most relevant regions of configuration space. We then demonstrate how baseline pair-potentials ("R6") can help to capture the long-range van der Waals (vdW) dispersion interactions that are important in black P[24] and other allotropes[26], and how this baseline can be combined with a shorter-ranged ML model—together allowing our model to learn from data at the DFT plus many-body dispersion (DFT + MBD) level of theory[59,60]. The new GAP (more specifically, GAP + R6) force field combines a transferable description of disordered, e.g., liquid P with previously unavailable accuracy in modelling the crystalline phases and their exfoliation. We therefore expect that this ML approach will enable a wide range of simulation studies in the future.

## Results

**A reference database for phosphorus.** The quality of any ML model depends on the quality of its input data. In the past, atomistic reference databases for GAP fitting have been developed either in a manual process (see, e.g., ref. [61]) or through GAP–RSS runs[62,63]—but these two approaches are inherently different, in many ways diametrically opposed, and it has not been fully clear what is the optimal way to combine them. We introduce here a reference database for P, which does indeed achieve the required generality, containing the results of 4798 single-point DFT + MBD computations, which range from small and highly symmetric unit cells to large supercell models of phosphorene. Of course, "large" in this context can mean no more than few hundred atoms per cell, which leads to one of the primary challenges in developing ML force fields: selecting properly sampled reference data to represent much more complex structures.

Whilst details of the database construction are given in Supplementary Note 1, we provide an overview by visualising its composition in Fig. 1. To understand the diversity of structures and the relationships between them, we use the smooth overlap of atomic positions (SOAP) similarity function[64,65]: we created a 2D map in which the distance between two points reflects their structural distance in high-dimensional space, here obtained from multidimensional scaling. In this 2D map, two SOAP kernels with cut-offs of 5 and 8 Å are linearly combined to capture short- and medium-range order. Every fifth entry of the database is included in the visualisation, for numerical efficiency.

Figure 1 allows us to identify several aspects of the constituent parts of the database. The GAP–RSS structures, taken from ref. [58], are indicated by grey points, and these are widely spread over the 2D space of the map: the initial randomised structures were generated using the same software (buildcell) as in the established Ab Initio Random Structure Searching (AIRSS) framework[66], with subsequent relaxations driven by evolving GAP models[58]. The purpose of including those data is to cover a large variety of different structures, with diversity being more important than accuracy. For the manually constructed part, in contrast, related structures cluster together, e.g., the various distorted unit cells representing white P (top left in Fig. 1). Melting white P leads to a low-density fluid in which $P_4$ units are found as well, and the corresponding points in the 2D visualisation are relatively close to those of the white crystalline form (marked as **1** in Fig. 1). Pressurising the low-density liquid leads to a liquid–liquid-phase transition (LLPT)[19–21], and accordingly points representing denser liquid structures are also

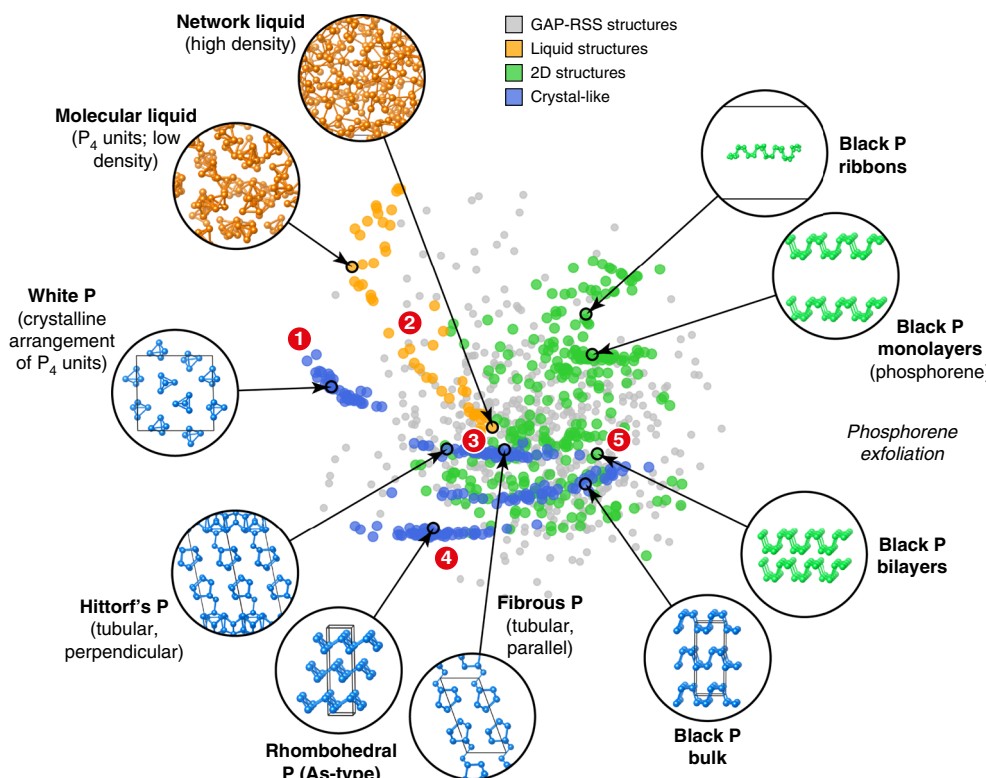

**Fig. 1 A GAP fitting database for elemental phosphorus.** The relationships between the structures in the database are visualised through 2D embedding of a SOAP similarity metric. Example structures are shown, and specific points of interest are highlighted by numbers: the closeness between molecular crystalline (white P, **1**) and liquid $P_4$, the transition between the molecular and network liquid (**2**), the similarity between Hittorf's and fibrous P, which both consist of extended tubes and fall in the same island on the plot (**3**), an isolated set of points corresponding to As-type structures (**4**) and the exfoliation from black P into bilayers (**5**) and monolayers. The GAP–RSS dataset from ref. [58], finally, is shown using smaller grey points and spans a wide range of configurations (see also Supplementary Fig. 1). Note that this map does not include the isolated P, $P_2$ and $P_4$ configurations, as it aims to survey the space of extended P structures.

found closer to the centre of the map (the transition between them occurs in the region marked as **2**). The high-density liquid itself, remarkably, appears to be structurally rather similar to Hittorf's and fibrous P, and the latter two crystalline allotropes occupy the same cluster of points in Fig. 1 (**3**)—reflecting the fact that they are built up from very similar, cage-like units[10]. Rhombohedral (As-type) P is further away from other entries, in line with the fact that no such allotrope is stable at ambient pressure (**4**)[67]. Finally, the right-hand side of Fig. 1 prominently features points corresponding to various types of black P and phosphorene-derived structures (an example of a bilayer is marked as **5**).

The various parts of the database pose a challenge to the ML algorithm: it needs to achieve a highly accurate fit for the crystalline configurations (blue in Fig. 1), yet retain the ability to interpolate smoothly between liquid configurations (orange). In this, the selection of input data is intimately connected with the regression task itself. A key feature of our approach is the use of a set of expected errors (regularisation), which is required to avoid overfitting (a GAP fit without regularisation would perfectly reproduce the input data, but lead to uncontrolled errors for even slightly different atomistic configurations). We set these values manually, bearing in mind the physical nature of a given set of configurations[61]: e.g., we use a relatively large value for the highly disordered liquid structures (0.2 eV Å$^{-1}$ for forces), but a smaller value for the bulk crystals (0.03 eV Å$^{-1}$). Similarly, large expected errors for the initial GAP–RSS configurations allow the force field to be flexible in that region of configuration space[63]—thus

ensuring that it remains usable for crystal-structure prediction in the future, which constitutes a very active research field for P[15–18] and can be vastly accelerated by ML force fields[18,58]. Details of the composition of the database developed here and the regularisation are given in Supplementary Notes 1 and 2.

**GAP + R6 fitting.** The next task in development of our ML force field is the choice of structural descriptors. In the case of P, there is a need to accurately describe the long-range vdW interactions between phosphorene sheets or in the molecular liquid—which are weak on an absolute scale, yet crucial for stability and properties. At the same time, the ML model must correctly treat complex, short-ranged, covalent interactions, e.g., in Hittorf's P with its alternating $P_9$ and $P_8$ cages[9]; it is this length scale (5-Å cut-off) that is typically modelled by finite-range descriptors in ML force fields[49–51].

Figure 2a–c illustrates the combination of descriptors used to "machine-learn" our force field (details are provided in the "Methods" section). The baseline is a long-range (20-Å cut-off) interaction term as in ref. [68], here fitted to the DFT + MBD exfoliation curve of black P. The latter is taken to be indicative for vdW interactions in P allotropes more generally, and a test for the transferability of this approach to more complex structures (Hittorf's P) is given in one of the following sections. The baseline model is subtracted from the input data, and an ML model is fitted to the energy difference, which is itself composed of two terms: a pair potential and a many-body term, both at short range

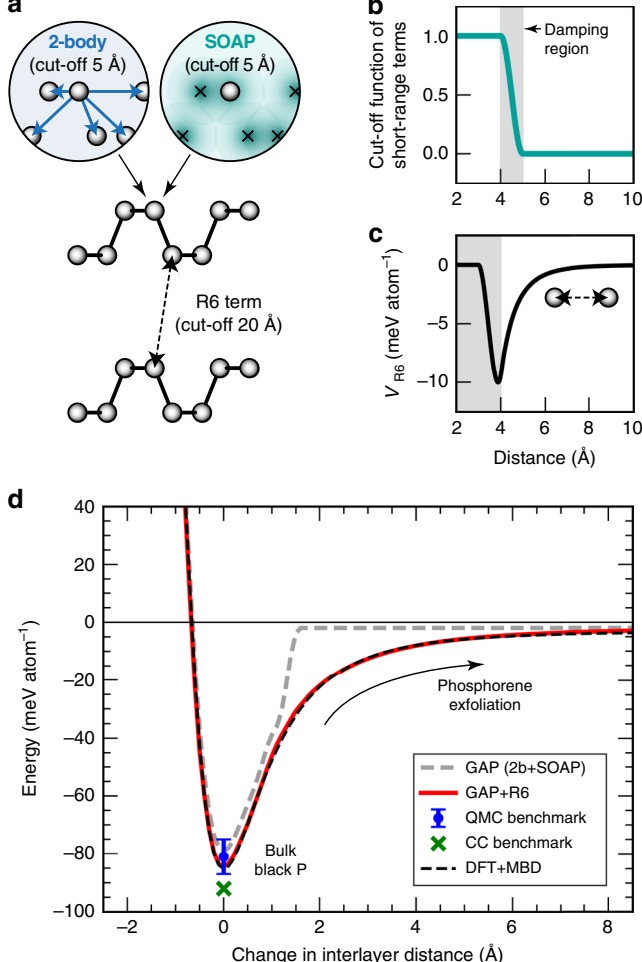

**Fig. 2 A GAP + R6 ML model including long-range dispersion.**
**a** Schematic sketch of the different types of structural descriptors, here illustrated for a pair of partially exfoliated phosphorene sheets—emphasising the medium-range (5 Å) and long-range (20 Å) descriptors that are combined in our approach ("Methods" section)[68]. **b**, **c** Modelling the different length scales: the upper panel shows the cut-off function used to bring the 2-body and SOAP descriptors smoothly to zero between 4 and 5 Å; the lower panel shows the long-range term, $V_{R6}$, evaluated for an isolated pair of atoms in the absence of the ML terms. **d** Phosphorene exfoliation curve from our GAP + R6 model (red) compared to the DFT + MBD reference (dashed black line), giving the energy computed for black P (structure from ref. [70]) as a function of the interlayer distance. A GAP fit without the long-range "+R6" term, i.e., based only on a 2b+SOAP fit with a 5-Å cut-off, is included for comparison (dashed grey line). To obtain these curves, the sheets have been shifted along the [010] direction without further relaxation, and the energy is referenced to that of a free monolayer. Benchmark results for the exfoliation energy from quantum Monte Carlo (QMC, −81 ± 6 meV/atom, with bars showing the error given by the authors, ref. [31]) and coupled-cluster (CC, −92 meV/atom, ref. [32]) studies are given by symbols, both plotted at the horizontal zero.

(5-Å cut-off, Fig. 2a, b), linearly combined and jointly determined during the fit[69]. The short-range GAP and the long-range baseline model are then added up to give the final model ("Methods" section). Because of the $1/r^6$ dependence of the long-range part, we refer to this approach as "GAP + R6" in the following.

Figure 2d shows the resulting exfoliation curve: we obtain it by scaling the known black P structure[70] in small steps along the [010] direction, keeping the individual puckered layers intact and

computing the potential energy at each step, with the energy of a free monolayer set as the energy zero. To illustrate the need for a treatment of long-range interactions (here, achieved using the "+R6" baseline), we fitted a GAP *without* this term, using a 5-Å cut-off and otherwise similar parameters—this model clearly fails to capture the longer-range interactions involved in the exfoliation, as shown by a grey dashed line in Fig. 2d. In contrast, the GAP + R6 result (red) and the DFT + MBD reference data (black) are practically indistinguishable. We also include two benchmark values from high-level quantum chemistry, one from quantum Monte Carlo computations[31], one from a coupled-cluster (CC) approach in ref. [32]. The GAP + R6 prediction (−85 meV per atom) is in excellent agreement with both, and it matches the DFT + MBD result to within 1% (≈0.8 meV). To place our results into context, we may quote from a recent study[27], which compared several computational approaches in regard to how well they describe the exfoliation energy of black P: the results varied widely, from about −10 meV (without any dispersion corrections) to between −86 and −145 meV (all at the PBE0 + D3 level but using different basis sets and damping schemes), and further to −218 meV for one specific combination of methods[27]. The same study provided initial evidence for the high performance of the MBD method in describing black P[27].

The most direct way to ascertain the quality of the ML model is to compute energies and forces for a separate test set of structures, and to compare the results to reference computations using DFT + MBD (the ground truth to be learned). We separate the results according to various types of test configurations, which are of a very different nature.

Figure 3a shows such tests for P structures obtained from GAP–RSS[58], starting with initial (random) seeds and progressively including more ordered and low-lying structures. The forces in the initial seeds range up to very high absolute values, as a response to atoms having been placed far away from local minima; the datapoints scatter but overall reveal a good correlation between DFT + MBD and GAP + R6. In contrast, Fig. 3b focuses on the manually constructed parts of our database: for the network liquid, there is still notable scatter, but for the molecular liquid and especially for the 2D and crystalline structures, the errors are much smaller. This is expected as these configurations correspond to distorted copies of only a few crystalline structures that are abundantly represented in the database. We emphasise that the test structures are not fully relaxed, on purpose (and neither are those used in the ML fit): they serve to sample slightly distorted environments where there are non-zero forces on atoms.

Numerical results for the test-set errors are given in Table 1. We emphasise that the initial (random) GAP–RSS configurations are included primarily for structural diversity, and that they experience very large absolute forces, ranging up to about 20 eV Å$^{-1}$ (Fig. 3a), much more than the test-set error. The much smaller magnitude of errors for the more ordered configurations is consistent with a progressively tightened regularisation of the GAP fit[61]: for example, we set the force regularisation to 0.4 eV Å$^{-1}$ for random GAP–RSS configurations, 0.2 eV Å$^{-1}$ for liquid P, but 0.03 eV Å$^{-1}$ for bulk crystalline configurations (Supplementary Table 1). The results for the subset describing the crystalline phases are in line with a recent benchmark study for six elemental systems, reporting energy RMS errors in the meV-per-atom region and force RMS errors from 0.01 eV Å$^{-1}$ (crystalline Li) to 0.16 eV Å$^{-1}$ (Mo) obtained from GAP fits[52]. Another recent test for liquid silicon showed errors of about 12 meV at.$^{-1}$ and 0.2 eV Å$^{-1}$ for energies and forces, respectively[71], which again is qualitatively consistent with our findings—the molecular liquid primarily consists of $P_4$ units, whereas the network liquid contains more diverse coordination numbers and

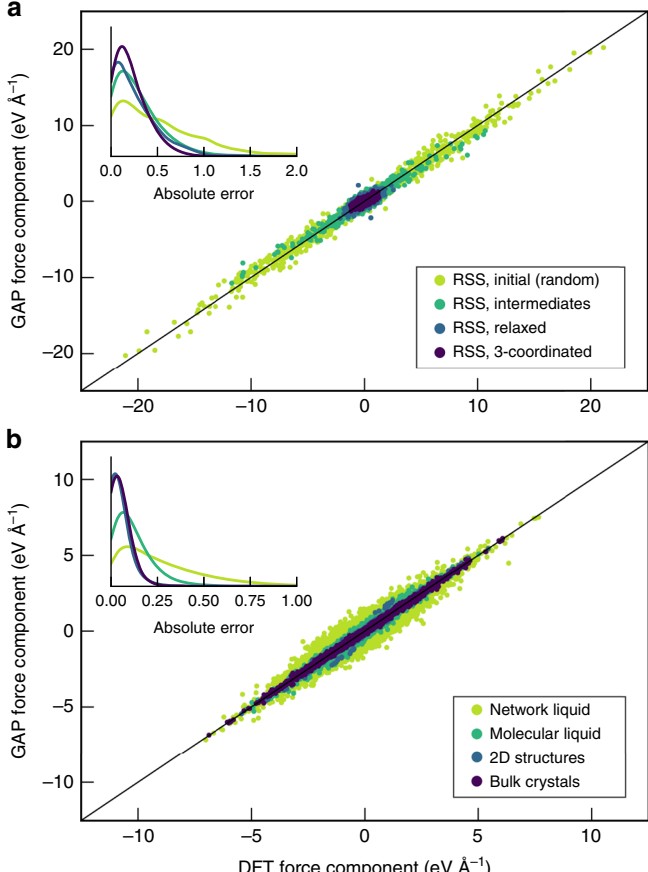

**Fig. 3 Validation of the ML force field.** Scatterplots of Cartesian force components for a test set of structures, which has not been included in the fit, comparing DFT + MBD computations with the prediction from GAP + R6. Data are shown for different sets of the GAP–RSS-generated (panel **a**) and manually constructed (panel **b**) parts of the database. The insets show kernel-density estimates ("smoothed histograms") of absolute errors with the same colour coding. Note the difference in absolute scales for the force components between the two panels.

**Table 1 Root mean square error (RMSE) measures for energies and force components[a].**

| | | RMSE energies (eV at.$^{-1}$) | RMSE forces (eV Å$^{-1}$) |
|---|---|---|---|
| GAP-RSS | Initial (random) | 0.116 | 0.69 |
| | Intermediates | 0.055 | 0.38 |
| | Relaxed | 0.058 | 0.36 |
| | 3-coordinated | 0.032 | 0.26 |
| Network liquid | | 0.008 | 0.36 |
| Molecular liquid | | 0.002 | 0.15 |
| 2D structures | | 0.002 | 0.07 |
| Bulk crystals | | 0.001 | 0.06 |

[a]The relevant parts of the database were randomly split into "training" and "testing" sets. The training data were collected, amended with additional (e.g., dimer) configurations and used as input for the ML fit. The testing data were not included in the fit, and RMSE errors are given for the latter, comparing our GAP + R6 model to DFT + MBD data. The total size of the training (testing) set is 4798 (1601) cells, respectively. Details are given in Supplementary Table 1.

environments, and its quantitative fitting error is therefore larger than that for its molecular counterpart (Table 1). We stress again that in the GAP framework, the ability to achieve good accuracy in one region of configuration space whilst retaining flexibility in

others depends strongly on the judicious choice of regularisation parameters (Supplementary Note 2 and Supplementary Table 2).

**Crystalline allotropes**. Phosphorus crystallises in diverse structures—and a substantial body of literature describes their synthesis and experimental characterisation. Among these crystalline allotropes, black P has been widely studied as the precursor to phosphorene. DFT + MBD describes the structure of bulk black P remarkably well[27], reproducing experimental data within any reasonable accuracy (Supplementary Note 3 and Supplementary Table 3). It is then, by extension, satisfying to observe the very high accuracy of the GAP + R6 prediction, which captures even the parameter $b$, corresponding to the interlayer direction, to within better than 0.5% of the DFT + MBD reference. The two inequivalent covalent bond lengths in black P, after full relaxation, are 2.225/2.255 Å (DFT + MBD) and 2.225/2.260 Å (GAP + R6), showing very good agreement.

Energies and unit-cell volumes of the main crystalline allotropes are given in Table 2. Strikingly, black, fibrous and Hittorf's P are essentially degenerate in their DFT + MBD ground-state energy, coming even closer together than an earlier study with pairwise dispersion corrections had indicated[26]. This de facto degeneracy is reproduced by our force field (Table 2), with all three structures being similar in energy to within 0.003 eV per atom. In terms of unit-cell volumes, black P is more compact, whereas fibrous and Hittorf's P contain more voluminous tubes and arrive at practically the same volume, as both contain the same repeat unit and only differ in how the tubes are oriented in the crystal structures. GAP + R6 reproduces all these volumes to within about 1%. White P, which we describe by the ordered β rather than the disordered α modification[2], is notably higher in energy, as expected for the highly reactive material. We finally include in Table 2 the rhombohedral As-type modification, which is a hypothetical structure at ambient conditions and can only be stabilised under pressure[72]. It is thus somewhat surprising that DFT + MBD assigns a slightly more negative energy to As-type than to black P (Table 2)—consequently, our ML model faithfully reproduces this feature, to within 0.002 eV per atom.

**Hittorf's phosphorus in 3D and 2D**. The exfoliation of black P to form phosphorene had already served as a case to illustrate the role of short-ranged versus GAP + R6 models (Fig. 2). Whilst most of the work on 2D phases is currently focused on phosphorene, Schusteritsch et al. suggested to exfoliate Hittorf's P to give "hittorfene"[73], and very recently Hittorf-based monolayers[11] and nanostructures[14] were indeed experimentally realised. It is therefore of interest to ask whether this exfoliation can be described by a force field for P, especially as the process involves more complex structures, making the routine application of DFT + MBD more computationally costly than for phosphorene. The exfoliation of Hittorf's P is also a more challenging test for our method: regarding black P, we had included multiple partially exfoliated mono- and bilayer structures in the database (Fig. 1), whereas for Hittorf's, we only include distorted variants of the experimentally reported bulk structure but no exfoliation snapshots or monolayers. Testing the ML force field on the full exfoliation curve therefore constitutes a more sensitive test for its usefulness in computational practice.

Figure 4 shows the exfoliation similar to Fig. 2d, but now for Hittorf's P, using two different structures. One is the initially reported refinement result by Thurn and Krebs (1969, purple in Fig. 4)[9]. The other was recently reported by Zhang et al. (2020, cyan)[11]. The samples in both studies have been synthesised in very different ways: the earlier study followed the original synthesis route by Hittorf[74], viz. slow cooling of a melt of white P

**Table 2 Unit-cell volumes and energies (relative to black P) for relevant allotropes, comparing DFT + MBD and GAP + R6 results.**

|  | Volume (Å³/atom) | | | | ΔEnergy (eV/atom) | |
| --- | --- | --- | --- | --- | --- | --- |
|  | Expt. | DFT + MBD | GAP + R6 | Error versus DFT (%) | DFT + MBD | GAP + R6 |
| White P ($\beta$-P$_4$, $P\bar{1}$) | 25.99[a] | 25.76 | 25.72 | −0.2 | +0.178 | +0.154 |
| Fibrous P ($P\bar{1}$) | 21.7[b] | 21.77 | 22.09 | +1.4 | +0.001 | +0.003 |
| Hittorf's P ($P2/c$) | 21.8[c] | 21.82 | 22.01 | +0.9 | −0.001 | ±0.000 |
| Black P ($Cmce$) | 19.03[d] | 18.83 | 18.74 | −0.5 | ±0 (reference) |  |
| As-type P ($R\bar{3}m$) | –[e] | 15.48 | 15.44 | −0.2 | −0.011 | −0.009 |

[a]From ref. 2; XRD at 88 K.
[b]From ref. 10; XRD at 293(2) K.
[c]Calc. from lattice parameters in ref. 9; XRD at room temperature.
[d]From ref. 80; XRD at 293 K.
[e]High-pressure allotrope[67], here studied in a hypothetical form without external pressure. A recent experimental study reports 14.6 Å³/atom at 6.05 GPa[72].

and excess Pb; the 2020 study used a chemical vapour transport

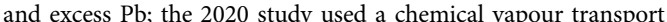

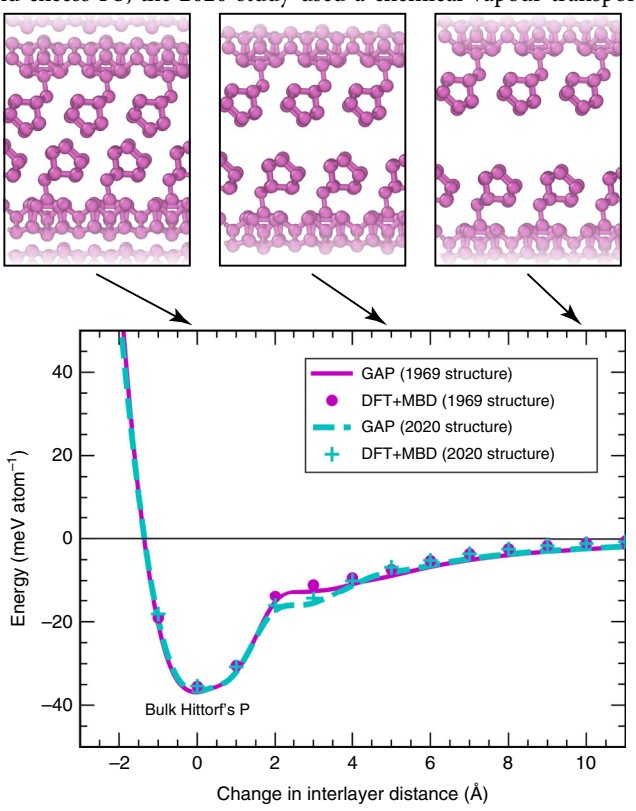

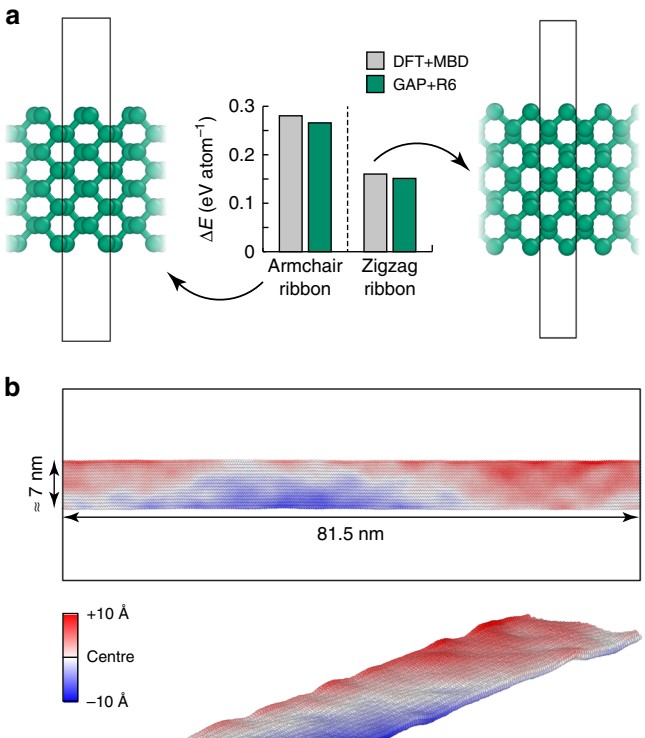

**Fig. 4 Exfoliation of Hittorf's phosphorus.** Exfoliation into monolayer "hittorfene"[73], similar to Fig. 2d, but now for a more complex structure where training data are only available around the minimum. Two different experimental structural models are used as a starting point: the initial $P2/c$ structure (1969, ref. 9, magenta), and a very recently proposed $P2/n$ structure (2020, ref. 11, cyan). The results of our GAP + R6 model are given by solid and dashed lines, respectively, and reference DFT + MBD computations are indicated by circles and crosses.

**Fig. 5 Phosphorene nanoribbons. a** The two fundamental types of ribbons, obtained by cleaving along the two in-plane directions of phosphorene, leading to armchair- (*left*) and zigzag- (*right*) type ribbons, with the boundaries of the periodic simulation cells indicated. The energies are given relative to a phosphorene monolayer; all structures are cleaved from the bulk without further relaxation. **b** Demonstration of the applicability to a much larger system (15,744 atoms), shown for a GAP + R6-driven MD snapshot after 10 ps in the NVT ensemble, with a thermostat set to 300 K, and then another 40 ps of constant-energy (NVE) dynamics. Colour coding indicates the atomic positions in the direction normal to the layer.

route[11], which may have led to slightly different ways in which the tubes are packed.

Remarkably, DFT + MBD places the two structures at practically degenerate exfoliation energies (about 35 meV/atom below the respective monolayer), without a discernible preference for one over the other, despite the different synthesis pathways and crystallographically dissimilar structure solutions[9,11]. Our ML force field fully recovers this degeneracy at around the

minimum (corresponding to the bulk phases) and at large interlayer spacing (above + 4 Å), as well as a subtle difference between the phases at intermediate separation. As pointed out by Schusteritsch et al.[73], the overall interlayer binding energy of Hittorf's P is very low, notably smaller than that of black P.

**Nanoribbons**. Akin to graphene nanoribbons, phosphorene can be cut into nanoribbons as well, as predicted computationally[75] and later demonstrated in experiment[76]. Such ribbons have been studied, e.g., in ref. [77], using empirical potential models. In Fig. 5a, we show the two fundamental types of phosphorene nanoribbons, referred to as "armchair" and "zigzag". The latter is clearly favoured among the two, and GAP + R6 reproduces the associated energetics to within 5–6% of the DFT + MBD result. The ratio between the formation energies of the armchair and zigzag ribbon, as the most important indicator for the stability preference, is even better reproduced, viz. 1.75 (DFT + MBD) compared to 1.76 (GAP + R6)[75].

The test in Fig. 5a assesses very small ribbons, because the effect of nanostructuring is most pronounced for those—in contrast, larger ribbons are more similar to 2D phosphorene, which is already ubiquitously represented in the database (Fig. 1). However, beyond this initial test, the ML force field brings substantially larger system sizes within reach. Figure 5b shows a zigzag phosphorene nanoribbon that is >80 nm in length, with a width that is consistent with experimental reports[76]. After a short NVT simulation, the system is allowed to evolve over 40 ps, leading to the visible formation of nanoscale ripples—each extending over several nanometres. This computational task may be compared with an earlier study using an empirical potential to simulate water diffusion on rippled graphene (over much longer timescales)[78]: with typical system sizes of $15 \times 15$ nm$^2$, and reaching up to $30 \times 30$ nm$^2$, such simulations are completely out of reach for quantum-mechanical methods, but they are accessible to ML force fields. Beyond the capability test in Fig. 5b, similar simulation cells, but with added heat sources and sinks, are widely used in computational studies of thermal transport, normally in combination with empirical potentials—as has indeed been shown for phosphorene nanoribbons[77]. The high accuracy of our ML model for predicting interatomic forces (0.07 eV Å$^{-1}$ for the 2D configurations, Table 1) allows one to anticipate a good performance for properties that are directly derived from the force constants, viz. phonon dispersions and thermal transport, as demonstrated previously for silicon (see refs. [61,71], and references therein). A rigorous study of phonons and thermal transport in phosphorene with GAP + R6 is envisioned for the future.

**Liquid phosphorus**. Liquid phases provide a highly suitable test case for the quality of a force field—indeed, the very first high-dimensional ML force field, an artificial neural-network model for silicon, was tested for the RDF of the liquid phase[42]. Phosphorus is, again, interesting in this regard, because two physically distinct phases and the occurrence of a first-order LLPT have been reported[19–21]. In Fig. 6, we validate our method for both phases, using simulation cells containing 248 atoms. The former (Fig. 6a–c) contains P$_4$ molecules; the latter (Fig. 6d–f) describes a covalently connected network liquid. We performed DFT-MD computations for reference; due to the high computational cost, these had to be carried out at the pairwise dispersion-corrected PBE + TS (rather than MBD) level[79]. Two different temperatures, 1000 and 2000 K, span the approximate temperature range in which phase transitions in P have been experimentally studied[20].

Our GAP + R6-driven MD simulations (which we call "GAP-MD" for brevity) describing the low-density molecular phase are in excellent agreement with the DFT-MD reference. The simplest structural fingerprint is the radial distribution function (RDF), plotted in Fig. 6b: there is a clearly defined first peak (corresponding to P–P bonds inside the P$_4$ units, with a maximum at about 2.2 Å) and, separated from it, an almost unstructured heap at larger distances beyond about 3 Å, all

indicative of a molecular liquid that consists of well-defined and isolated units. Similarly, the angular distribution functions (ADF) in Fig. 6c show a single peak at ≈60°, consistent with the equilateral triangles that make up the faces of the ideal P$_4$ molecule. The molecules are more diffusive at higher temperature, and therefore, the features in the radial and angular distributions are slightly broadened in the 2000-K data compared to those at 1000 K—but there are no qualitative changes between the two temperature settings, and the GAP-MD simulation reproduces all aspects of the DFT-MD reference.

In Fig. 6d–f, we report the same tests but now for the network liquid. In this case, at 1000 K, the GAP-MD-simulated liquid appears to be slightly more structured than that from DFT-MD, indicated by a larger magnitude of the second RDF peak between 3 and 4 Å, and a somewhat sharper peak in the angular distribution at about 100° in the GAP-MD data. Whether that is a significant difference between DFT and GAP + R6 or merely a consequence of the slightly different dispersion treatments, MD algorithm implementations, etc. remains to be seen—but it does not change the general outcome that all major features of the DFT-based trajectory are well reproduced by the GAP + R6 model. The 2000-K structures generated by DFT-MD and GAP-MD simulations agree very well with each other, likely within the expected uncertainty that is due to finite-system sizes and simulation times. A feature of note in the ADF is a secondary peak at 60°, much smaller than in the molecular liquid (Fig. 6c), but present nonetheless: the liquid, especially at higher temperature, does still contain three-membered ring environments. Comparing the 1000- and 2000-K simulations, the former reveals a clear predominance of bond angles between about 90° and 110°, whereas the bond-angle distribution in the latter is much more spread out, indicating a highly disordered liquid structure.

**Liquid–liquid-phase transition**. We finally carried out a simulation of the LLPT, expanding substantially on prior DFT-based work[33–36] in terms of system size, as shown in Fig. 7. Our initial system contains 496 thermally randomised P$_4$ molecules (1984 atoms in total), which are initially held at the 2000-K and 0.3-GPa state point for 25 ps. We then compress the system with a linear-pressure ramp to 1.5 GPa, over a simulation time of 100 ps. At low densities, the system consists entirely of P$_4$ units, most having distorted tetrahedral shapes (and thus threefold coordination, indicated by light-blue colouring in Fig. 7a). Occasionally during the high-temperature dynamics, tetrahedra open up such that two atoms temporarily lose contact and thus have lower coordination numbers; sometimes two tetrahedra come closer than the distance we use to define bonded contacts (2.7 Å, as in Fig. 6b). All these effects are minor, as seen on the left-hand side of Fig. 7a. Upon compression, the atomistic structure changes drastically: having reached a pressure of 0.81 GPa, the system has transformed into a disordered, covalently bonded network, qualitatively consistent with previous simulations in much smaller unit cells[33–36], but now providing insight for a system size that would have been inaccessible to DFT-MD simulations at this level. To benchmark the computational performance of GAP-MD, we repeated this simulation using 288 cores on the UK national supercomputer, ARCHER, where it required 6 h (corresponding to 0.5 ns of MD per day). The LLPT gives rise to a much larger diversity of atomic coordination environments, seen on the right-hand side of Fig. 7a. We emphasise that the liquid is held at a very high temperature of 2000 K, and therefore substantial deviations from the ideal threefold coordination (that would be found in crystalline P) are to be expected.

We analyse this GAP-MD simulation in Fig. 7b. We first record the density of the system as a function of applied pressure.

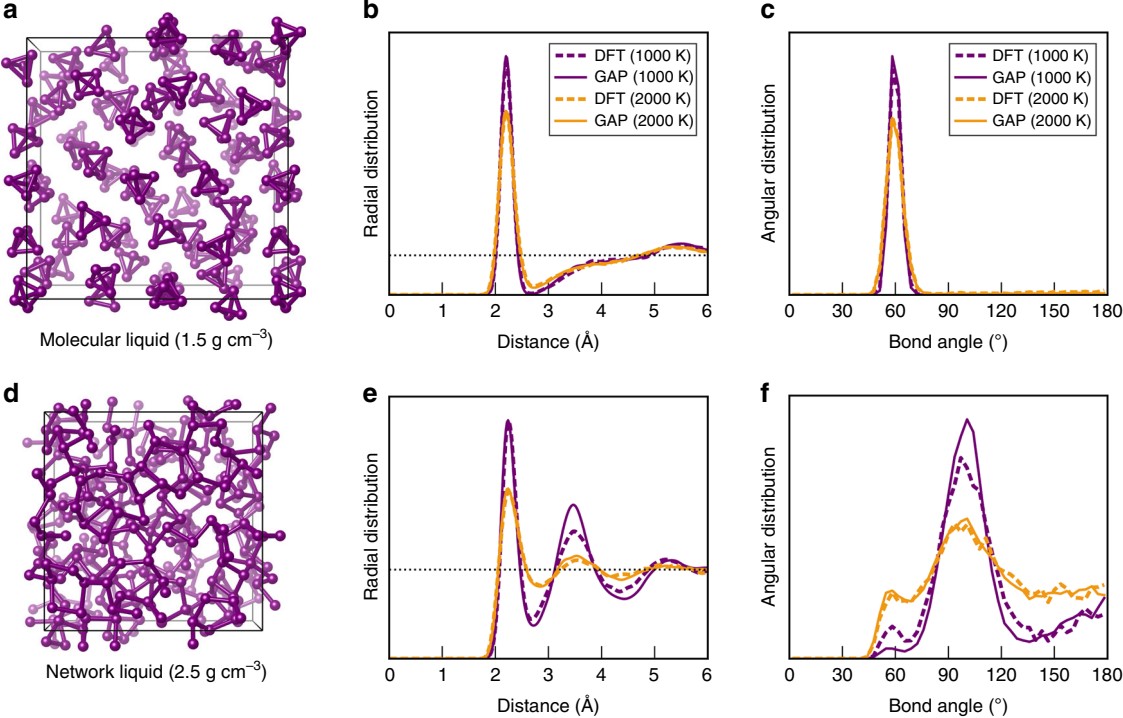

**Fig. 6 Liquid phosphorus.** MD simulations in the NVT ensemble, benchmarking the quality of the GAP + R6 ML force field for the description of liquid phosphorus. **a** Snapshot of a DFT-MD simulation of a system containing 62 $P_4$ molecules at a fixed density of 1.5 g cm$^{-3}$, corresponding to the low-density liquid (or fluid). **b** Radial distribution functions for this system at two different temperatures, taken from the last 10 ps of the trajectories. Solid lines indicate GAP-MD simulations, whereas dashed lines show the results of reference DFT-MD trajectories. **c** Same for angular distribution functions (ADF), computed using a radial cut-off of 2.7 Å. **d–f** Same but for the network liquid at a much higher density of 2.5 g cm$^{-3}$. The slightly more "jagged" appearance of the DFT data in panel **f** is due to the smaller number of structures that are sampled from the trajectory.

The molecular liquid is quite compressible, indicated by a density increase of about 40% during compression from 0.3 to 0.7 GPa, consistent with the presence of only dispersive interactions between the molecules. When the system is compressed further, between 0.7 and 0.8 GPa, the density increases rapidly, concomitant with the observation of the LLPT in our simulation (Fig. 7a). The network liquid is much less compressible, and it is predicted to have a density of about 2.6–2.7 g cm$^{-3}$—very similar to the crystallographic density of black P (2.7 g cm$^{-3}$ at atmospheric pressure)[80], and smaller than 3.5 g cm$^{-3}$ reported for As-type P at about 6 GPa[72], in line with expectations. The transition, in fact, begins to occur earlier in the trajectory, as seen by analysing the count of threefold coordinated atoms and three-membered rings (the latter being a structural signature of the $P_4$ molecules). Coexistence simulations and thermodynamic integration are now planned to map out the high-temperature/high-pressure LLPT in comparison to experimental data[20].

## Discussion

We have developed a general-purpose ML force field for atomistic simulations of bulk and nanostructured forms of phosphorus, one of the structurally most complex elemental systems. Our study showed how a largely automatically generated GAP–RSS database can be suitably extended based on chemical understanding (in the ML jargon, "domain knowledge") whenever a highly accurate description of specific materials properties is sought. The present work might therefore serve as a blueprint for how general reference databases for GAP, and in fact other types of ML force fields for materials, can be constructed. In the present case, for example, reference data for layered (phosphorene) structures were added as well as for the LLPT, and our tests suggest the

resulting force field to be suitable for simulations of all these practically relevant scenarios. Proof-of-concept simulations were presented for a large (>80-nm-long) phosphorene nanoribbon, as well as for the liquid phases, showcasing the ability of ML-driven simulations to tackle questions that are out of reach for even the fastest DFT codes. Future work will include a more detailed simulation study of the liquid phases, as well as new investigations of red (amorphous) P, now all carried out at the DFT + MBD level of quality and with access to tens of thousands of atoms in the simulation cells. We certainly expect that phosphorus will continue to remain exciting, in the words of a recent highlight article[1]. We also expect that the approaches described here will be beneficial for the modelling of other systems with complex structural chemistry—including, but not limited to, other 2D materials that are amenable to exfoliation and could be described by GAP + R6 models in the future.

## Methods

**Reference data.** Dispersion-corrected DFT reference data were obtained at two different levels. Initially, we used the pairwise Tkatchenko–Scheffler (TS) correction[79] to the Perdew–Burke–Ernzerhof (PBE) functional[81], as implemented in CASTEP 8.0[82]. For the final dataset, we employed the MBD approach[59,60]. We expect that a similar "upgrading" of existing fitting databases with new data at higher levels of theory will be useful in the future, especially as higher levels of computational methods are coming progressively within reach (cf. the emergence of high-level reference computations for black P[31,32]), as has indeed been shown in the field of molecular ML potentials (see, e.g., ref. [83]). PBE + MBD data were computed using the projector-augmented wave method[84] as implemented in VASP[85,86]. The cut-off energy for plane waves was 500 eV; the criterion to break the SCF loop was a $10^{-8}$-eV energy threshold. Computations were carried out in spin-restricted mode. We used Γ-point calculations and real-space projectors (LREAL = Auto) for the large supercells representing liquid and amorphous structures; the remainder of the computations was carried out with automatic $k$-mesh generators with $l = 30$, where $l$ is a parameter that determines the number of divisions along each reciprocal lattice vector.

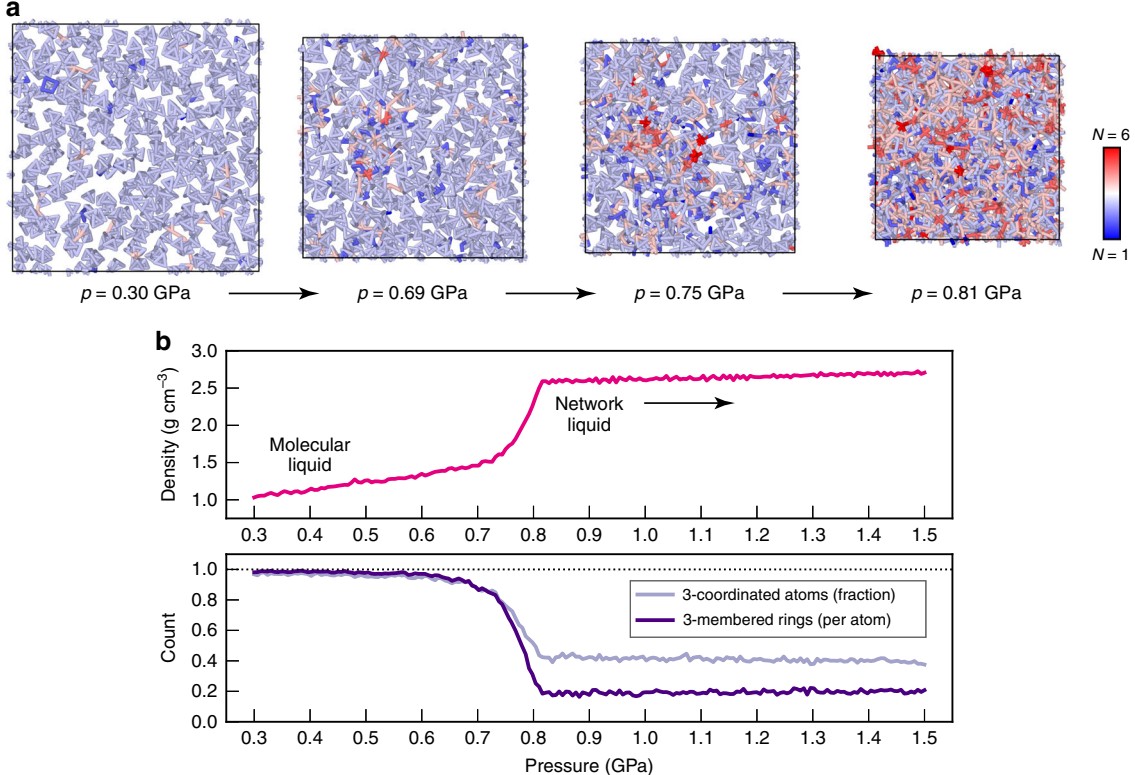

**Fig. 7 The liquid–liquid-phase transition.** We report a GAP-MD simulation of the liquid–liquid-phase transition (LLPT) in phosphorus, described by compressing a system of 1984 atoms from 0.3 to 1.5 GPa over 100 ps ($10^5$ timesteps), with the temperature set to 2000 K. **a** Consecutive snapshots from the trajectory, with coordination numbers, $N$, indicated by colour coding. **b** Evolution of density and atomic connectivity through the LLPT. The former, shown in the upper panel, starting with a low-density, compressible molecular liquid, increases rapidly between about 0.7 and 0.8 GPa, and then reaches much higher densities for the less compressible network liquid. The fraction of 3-coordinated atoms is unity in ideal $P_4$, but strongly lowered because of the LLPT; the network liquid contains much higher- and some lower-coordinated environments (cf. panel **a**). The count of three-membered rings can similarly be taken as an indicator for the presence of molecular $P_4$ units: in the ideal molecular liquid, only $P_4$ tetrahedra are found (each having four faces, and hence four three-membered rings, one per atom); in the network liquid, three-membered rings are still present, but their count is reduced to about a fifth, making way for larger rings consistent with a covalently bonded network.

**GAP + R6 fitting.** The GAP + R6 force field combines short-range ML terms and a long-range baseline (Fig. 2a) as follows. We start by fitting a Lennard–Jones (LJ) potential to the DFT + MBD exfoliation curve of black P at interatomic distances between 4 and 20 Å. We then define a cubic spline model, denoted $V_{R6}$, using the same idea as in ref. [68]. The baseline is described by a cubic spline fit that comprises the point (3.0 Å, 0 eV) together with the LJ potential between 4.0 and 20 Å, using spline points at 0.1-Å spacing up to 4.5 Å, and 0.5-Å spacing beyond that. The derivative of the potential is brought to zero at 3.0 and 20 Å; its shape is plotted in Fig. 2c. The fitted LJ parameters for our model are $\epsilon_6 = 6.2192$ eV; $\epsilon_{12} = 0$ (i.e., only the attractive longer-range part of the LJ potential is used); $\sigma = 1.52128$ Å. The baseline model is subtracted from the input data, and an ML model is constructed by fitting to

$$\Delta E = E_{DFT+MBD} - \sum_{i>j} V_{R6}(r_{ij}), \qquad (1)$$

where we denote the long-range potential by $V_{R6}$ for simplicity (because of its $1/R^6$ term), and $i$ and $j$ are atomic indices. The final model for the machine-learned energy of a given atom, $\varepsilon(i)$, thus reads

$$\varepsilon(i) = \left\{ \delta^{(2b)} \sum_q \varepsilon_i^{(2b)}(q) + \delta^{(MB)} \sum_{\mathbf{q}'} \varepsilon_i^{(MB)}(\mathbf{q}') \right\} + \frac{1}{2} \sum_j V_{R6}(r_{ij}). \qquad (2)$$

The first two sums in Eq. (2) together constitute the GAP model, combined using a properly scaled linear combination with scaling factors, $\delta$ (which are here given as dimensionless), and the last term, $V_{R6}$, is added to the ML prediction to give the final result. The two-body ("2b") and many-body (Smooth Overlap of Atomic Positions, SOAP[64]) models are defined by the respective descriptor terms: $q$ is a simple distance between atoms, which enters a squared-exponential kernel, and $\mathbf{q}'$ is the power-spectrum vector constructed from the SOAP expansion coefficients for the atomic neighbour density[64]. The ML fit itself is carried out using sparse Gaussian process regression as implemented in the GAP framework[43], employing a sparsification procedure that includes 15 representative points for the two-body descriptor and 8000 for SOAP. The full descriptor string used in the GAP

fit is provided in Listing 1, and together with the data and their associated regularisation parameters (Supplementary Notes 1 and 2), it defines the required input for the model. The potential is described by an XML file (see "Data availability" and "Code availability" statements).

**MD simulations.** DFT-MD simulations were done with VASP[85,86], using the pairwise TS correction for dispersion interactions[79] and an integration timestep of 2 fs. GAP-MD simulations were carried out with LAMMPS[87], either at constant volume for comparison with the DFT data (Fig. 6), or using a built-in barostat for pressurisation simulations (Fig. 7)[88–90]. The timestep in all GAP-MD simulations was 1 fs, which was found to improve the quality of the simulations compared to a 2-fs timestep. Whether this is a consequence of the somewhat different thermostats and MD implementations or, in fact, a consequence of the shape of the potential remains to be investigated—for the time being, we are content with running all GAP-MD simulations at the (more computationally costly) timestep of 1 fs.

**Listing 1: definition of the descriptor string used in the GAP fit.**
```
gap={distance_Nb order=2 cutoff=5.0 n_sparse=
15 covariance_type=ard_se delta=2.0 theta_uni
form=2.5 sparse_method=uniform compact_clus
ters=T: soap l_max=6 n_max=12 atom_sigma=0.5
cutoff=5.0 radial_scaling=0.5 cutoff_transi
tion_width=1.0 central_weight=1.0 n_sparse=
8000 delta=0.2 f0=0.0 covariance_type=dot_pro
duct zeta=4 sparse_method=cur_points}.
```

**Data availability**
The potential model described herein as well as the DFT+MBD reference data used for fitting the model are openly available through the Zenodo repository (https://doi.org/10.5281/zenodo.4003703). The unique identifier of the potential is GAP_2020_5_23_

`60_1_23_12_19`. In addition, the (DFT+MBD-computed) testing data used in this paper are available at https://github.com/libAtoms/testing-framework/tree/public/tests/P/.

## Code availability

The GAP code, which was used to carry out the fitting of the potential and the validation shown throughout this work, is freely available at https://www.libatoms.org/ for non-commercial research. The interface to LAMMPS (allowing GAPs to be used through a `pair_style` definition) is provided by the QUIP code, which is freely available at https://github.com/libAtoms/QUIP/.

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

## Acknowledgements

We thank N. Bernstein and J.R. Kermode for developing substantial parts of the potential testing framework (described in ref. [61]), which we have used in the present work. V.L.D. thanks C.J. Pickard and D.M. Proserpio for ongoing valuable discussions and the Leverhulme Trust for an Early Career Fellowship. Parts of this work were carried out during V.L.D.'s previous affiliation with the University of Cambridge (until August 2019) with additional support from the Isaac Newton Trust. V.L.D. and M.A.C. acknowledge travel support from the HPC-Europa3 initiative (in the framework of the European Union's Horizon 2020 research and innovation programme, Grant Agreement 730897). M.A.C. acknowledges personal funding from the Academy of Finland (grant number #310574) and computational resources from CSC—IT Center for Science. This work used the ARCHER UK National Supercomputing Service through EPSRC grant EP/P022596/1. The authors would like to acknowledge the use of the University of Oxford Advanced Research Computing (ARC) facility in carrying out this work (https://doi.org/10.5281/zenodo.22558). Post processing and visualisation of structural data was made possible by the freely available ASE[91], VESTA[92] and OVITO[93] software.

## Author contributions

V.L.D. initiated and coordinated the study. V.L.D. developed the reference database and fitted initial potential versions at the PBE+TS level; M.A.C. performed and analysed the reference computations at the PBE+MBD level; G.C. fitted the final potential version, including the long-range baseline. V.L.D. and G.C. jointly analysed and validated the potential. V.L.D. studied the liquid phases. V.L.D. wrote the paper with input from all authors.

## Competing interests

G.C. is listed as inventor on a patent filed by Cambridge Enterprise Ltd. related to SOAP and GAP (US patent 8843509, filed on 5 June 2009 and published on 23 September 2014). The remaining authors declare no competing interests.
