## [Peer Review File · Nature Communications]

REVIEWERS' COMMENTS:

Reviewer #1 (Remarks to the Author):

I read with interest the manuscript "A general-purpose machine-learning force field for bulk and nanostructured phosphorus" by Deringer, Caro and Csanyi. The noteworthy result of this article is the development and validation of a machine-learning force field for phosphorus. It also showcases applications of the model to the study of the liquid-liquid transition and a phosphorene nanoribbon. From the methodological point of view it seems to me that little is added with respect to existing literature, but in order to develop such a force field the authors had to harness many state of the art techniques (DFT calculations at the PBE+MBD level, GAP potentials with long range vdW interactions, Random structure searching). The development of this kind of force field is an important direction of research nowadays and this manuscript describes careful and thorough work that, in my opinion, pushes the field of molecular simulations forward.

This work will probably stimulate more research in the future both from the point of view of force field development and the application of this model to study other phenomena. Recently, similar papers that develop machine learning models for other systems have been published and they have garnered considerable attention.

The data shown in the paper supports the conclusions and the technical details seem to be correct. As far as I can see, enough details are provided to reproduce the work and the fact that the authors make the data and software available guarantees the reproducibility. The manuscript is very well written and accessible for people working in related fields. This work definitely meets the standards of quality in the field and probably exceeds them.

My recommendation is to accept this article after the authors consider the suggestions that I mention below.

1) I assume the regularisation described in page 7 was needed in order to achieve low errors in energies and forces of configurations in the most important regions of configuration space (bulk and 2 structures). The authors justify this procedure with the argument that it is hard to interpolate between "rapidly fluctuating liquid configurations". However I suspect that the GAP+R6 might lack the flexibility to fit all these configurations with low error. How did the GAP+R6 perform without the regularisation? If the performance was poor, I suggest that you state it and then describe the regularisation as a solution to this problem. GAP is a very useful and established framework and I think it would be useful to clarify this point.

2) From the plots in Figure 3 it is hard to understand how large the deviations of the forces are. I suggest that you include histograms of the errors in the forces so it is easier to interpret their magnitude and distribution.

3) I suggest that the authors report the performance of the GAP-MD simulations in ns/day or ps/day. The trajectories of the liquid liquid transition and the phosphorene nanoribbon are rather short ~ 100 ps and make me wonder if the wall time in these simulations is 1 day or 2 months. In the latter case the potential would not be very useful for real applications.

4) In the abstract the authors mention that the force field was trained on "highly converged data"? I fear that this is a claim that might not be substantiated. What do the authors mean? Perhaps some convergence of the DFT error below some threshold? If this is the case, does it make sense to state it considering that the errors in some regions of configuration space are rather large? My suggestion is to drop the "highly converged" from the phrase unless this point is clarified.

Dr. Pablo Piaggi
Postdoctoral Research Fellow
Department of Chemistry
Princeton University

Reviewer #2 (Remarks to the Author):

This paper presents a machine learning manybody classical potential energy surface for multiple phases and allotropes of phosphorous. This is a prominent result given the complexity of phosphorous interactions, which show a potent mix of covalent bonding and non-bonded London forces, all of which are sensitive to the local arrangement of the atoms in space. Prior to this work, atomistic simulations of phosphorous fell into two distinct classes: small-scale high-accuracy quantum electronic structure calculations based on Density Functional Theory or large-scale calculations based on empirical potential energy surfaces. The latter can be accurate in particular regions of configuration space (e.g. particular allotropes) but typically fail to accurately represent the energies of diverse structures, including liquid states and multiple allotropes. The former are accurate over a diverse range of configurations, but become computationally intractable beyond a few hundred atoms, too small to observe many of the collective rearrangements of phosphorous that drive the interesting macroscopically observable behaviors, including the formation of 2D materials, as well as the existence of multiple liquid states at relatively low pressures.

Through careful application of a hybrid approach for generating training data (random structure generation augmented by targeted specific phases and structural motifs), as well as a well-established approach to generating local structural fingerprints (descriptors), the authors have constructed a potential energy model that combines the accuracy of the quantum methods for diverse structures with the scalability of an empirical potential. The efficacy of this approach is demonstrated first by comparison with the diverse training data e.g. white, fibrous, Hittorf's and black phosphorous allotropes. This is further reinforced by validation of selected results against larger-scale quantum calculations, including calculations of liquid structure (rdf, adf) for two distinct liquid phases, as well as puckered phosphorene nanoribbons. Finally, the authors have used the new potential to directly simulate the liquid-liquid phase transition under compression, in which the low density molecular liquid of P₄ molecules switches to a high-density network liquid.

This diversity of structures captured by a single potential energy model is unprecedented in the rapidly evolving field of machine-learning interatomic potentials, and paves the way for more systematic explorations of phosphorous polymorphs, as well as many other exotic materials.

The paper is technically sound and carefully written. I have no suggestions for improvement.

Reviewer #3 (Remarks to the Author):

This work focuses on the fitting of a classical force field (FF) for elemental phosphorus in its bulk and nano-structures. The force field is developed using a machine-learning (ML) approach, so the development of a large database of structure to fit on is necessary. The authors clearly put a lot of effort in developing such a database, and the fact that the testing data is available to the public is greatly appreciated. I'm not clear if only the validation data is available, or also the data used for the ML fitting, so that should be made clearer. Obviously, it would be appreciated if the whole database would be available to the public.

The paper covers an important topic and provides an improved tool to the computational community, to facilitate the investigation of such an elusive, and allotropic-rich, element. This work is well written, clear, and extensively validated. It is worth publishing, if the author can provide a few extra clarifications/details. The ML-part of the work needs more details. While I understand it relies on previous publications, the paper should provide a self-contained description of the ML model used, parameters fitted and specifics on how the reported RMSE were obtained (80-10 split?, 90-10?, cross validation? etc.), even if only in the SI. Also, explicitly telling the number of validation structures versus fitting ones would be interesting. As the forces in Figure 3 are large with respect to what is considered necessary for structural convergence, it is not immediate to translate these results into force-field predictability. It would be more useful to show the same scatterplot type of comparison between the DFT formation energies of the validation structures and the corresponding GAP energies, once the initial structures are allowed to relax using the newly developed force field.

Lastly, it is common to use force fields to investigate temperature-dependent phenomena. While this work addresses temperature effects in the case of the liquid, it doesn't discuss how well this new force field performs for solids. For instance, thermal conductivity of black phosphorus nanosheets is of interest. Is this FF a good tool for such a study or not? These types of applications should be discussed.

Response to Reviewers' Comments – Manuscript NCOMMS-20-28827

We thank all three reviewers for their constructive and helpful comments. Below, we quote their reports in full; our point-by-point response is interspersed in **blue**, and action taken is detailed in **red**.

Reviewer #1 (Remarks to the Author):

I read with interest the manuscript "A general-purpose machine-learning force field for bulk and nanostructured phosphorus" by Deringer, Caro and Csanyi. The noteworthy result of this article is the development and validation of a machine-learning force field for phosphorus. It also showcases applications of the model to the study of the liquid-liquid transition and a phosphorene nanoribbon. From the methodological point of view it seems to me that little is added with respect to existing literature, but in order to develop such a force field the authors had to harness many state of the art techniques (DFT calculations at the PBE+MBD level, GAP potentials with long range vdW interactions, Random structure searching). The development of this kind of force field is an important direction of research nowadays and this manuscript describes careful and thorough work that, in my opinion, pushes the field of molecular simulations forward.

This work will probably stimulate more research in the future both from the point of view of force field development and the application of this model to study other phenomena. Recently, similar papers that develop machine learning models for other systems have been published and they have garnered considerable attention.

The data shown in the paper supports the conclusions and the technical details seem to be correct. As far as I can see, enough details are provided to reproduce the work and the fact that the authors make the data and software available guarantees the reproducibility. The manuscript is very well written and accessible for people working in related fields. This work definitely meets the standards of quality in the field and probably exceeds them.

Response: Thank you very much.

My recommendation is to accept this article after the authors consider the suggestions that I mention below.

1) I assume the regularisation described in page 7 was needed in order to achieve low errors in energies and forces of configurations in the most important regions of configuration space (bulk and 2 structures). The authors justify this procedure with the argument that it is hard to interpolate between "rapidly fluctuating liquid configurations". However I suspect that the GAP+R6 might lack the flexibility to fit all these configurations with low error. How did the GAP+R6 perform without the regularisation? If the performance was poor, I suggest that you state it and then describe the regularisation as a solution to this problem. GAP is a very useful and established framework and I think it would be useful to clarify this point.

Response: As the reviewer correctly states, the regularisation is a key component of the approach, because it allows us to “tighten” the fit for configurations where this is needed. In fact, without any regularisation, the method would not be usable in practice: it would perfectly fit the input data but lead to uncontrolled errors even for slightly different atomistic configurations (“overfitting”).

An interesting test, inspired by the reviewer’s suggestion, is to fit a version of the potential where no *custom* regularisation is used (small for crystalline phases, large for random structures, *etc.*), but instead the same regularisation is used throughout, keeping all other parameters of the fit unchanged. We performed such tests and found that, indeed, the quality of such a potential is worse for the crystalline phases: *e.g.*, with RMS energy and force errors of {0.006 eV, 0.16 eV Å⁻¹} for a uniform large expected error, compared to the result of the custom regularisation of {0.001 eV, 0.06 eV Å⁻¹}.

Action taken: We now state more clearly that the regularisation “is required to avoid overfitting (a GAP fit without regularisation would perfectly reproduce the input data, but lead to uncontrolled errors for even slightly different atomistic configurations)” (p. 8).

We added a more detailed discussion to the Supplementary Information, including new data for more loosely or more tightly regularised fits that do not adapt to different classes of input data, demonstrating that this leads to a detriment in performance for the 2D and crystalline phases (new Supplementary Note 2).

2) From the plots in Figure 3 it is hard to understand how large the deviations of the forces are. I suggest that you include histograms of the errors in the forces so it is easier to interpret their magnitude and distribution.

Response: This is a good suggestion.

Action taken: We added kernel density estimates (“smoothed histograms”) for the different parts of the dataset as insets to Figure 3.

3) I suggest that the authors report the performance of the GAP-MD simulations in ns/day or ps/day. The trajectories of the liquid liquid transition and the phosphorene nanoribbon are rather short ~ 100 ps and make me wonder if the wall time in these simulations is 1 day or 2 months. In the latter case the potential would not be very useful for real applications.

Response: This is a very important point that we should clarify. The simulation of the liquid–liquid transition, encompassing 125 ps of simulation time (125,000 timesteps), took about 40 hours on a local computing environment. In response to the reviewer’s comment, to obtain more widely comparable “benchmark” data, we repeated the simulation on the ARCHER UK national supercomputing service (Cray XC30) where it took 5:55:12 (h:min:sec) on 288 cores. With such low runtime requirements, we expect that the potential will indeed be useful for real-world applications.

Action taken: We now state on p. 23 that “To benchmark the computational performance of GAP-MD, we repeated this simulation using 288 cores on the UK national supercomputer, ARCHER, where it required 6 hours (corresponding to 0.5 ns of MD per day)”.

4) In the abstract the authors mention that the force field was trained on "highly converged data"? I fear that this is a claim that might not be substantiated. What do the authors mean? Perhaps some convergence of the DFT error below some threshold? If this is the case, does it make sense to state it considering that the errors in some regions of configuration space are rather large? My suggestion is to drop the "highly converged" from the phrase unless this point is clarified.

Response: Good point. We mean that dense k -point grids are used for the DFT single-point computations, which makes them “highly converged” in terms of noise in energies and forces. We agree, however, that this would need clarification (which would detract from the main message of the abstract).

Action taken: We removed the phrase “highly converged” from the abstract, as suggested.

Dr. Pablo Piaggi
Postdoctoral Research Fellow
Department of Chemistry
Princeton University

Reviewer #2 (Remarks to the Author):

This paper presents a machine learning manybody classical potential energy surface for multiple phases and allotropes of phosphorous. This is a prominent result given the complexity of phosphorous interactions, which show a potent mix of covalent bonding and non-bonded London forces, all of which are sensitive to the local arrangement of the atoms in space. Prior to this work, atomistic simulations of phosphorous fell into two distinct classes: small-scale high-accuracy quantum electronic structure calculations based on Density Functional Theory or large-scale calculations based on empirical potential energy surfaces. The latter can be accurate in particular regions of configuration space (e.g. particular allotropes) but typically fail to accurately represent the energies of diverse structures, including liquid states and multiple allotropes. The former are accurate over a diverse range of configurations, but become computationally intractable beyond a few hundred atoms, too small to observe many of the collective rearrangements of phosphorous that drive the interesting macroscopically observable behaviors, including the formation of 2D materials, as well as the existence of multiple liquid states at relatively low pressures.

Through careful application of a hybrid approach for generating training data (random structure generation augmented by targeted specific phases and structural motifs), as well as a well-established approach to generating local structural fingerprints (descriptors), the authors have constructed a potential energy model that combines the accuracy of the quantum methods for diverse structures with the scalability of an empirical potential. The efficacy of this approach is demonstrated first by comparison with the diverse training data e.g. white, fibrous, Hittorf's and black phosphorous allotropes. This is further reinforced by validation of selected results against larger-scale quantum calculations, including calculations of liquid structure (rdf, adf) for two distinct liquid phases, as well as puckered phosphorene nanoribbons. Finally, the authors have used the new potential to directly simulate the liquid-liquid phase transition under compression, in which the low density molecular liquid of P₄ molecules switches to a high-density network liquid.

This diversity of structures captured by a single potential energy model is unprecedented in the rapidly evolving field of machine-learning interatomic potentials, and paves the way for more systematic explorations of phosphorous polymorphs, as well as many other exotic materials.

The paper is technically sound and carefully written. I have no suggestions for improvement.

Response: Thank you very much.

Reviewer #3 (Remarks to the Author):

This work focuses on the fitting of a classical force field (FF) for elemental phosphorus in its bulk and nano-structures. The force field is developed using a machine-learning (ML) approach, so the development of a large database of structure to fit on is necessary. The authors clearly put a lot of effort in developing such a database, and the fact that the testing data is available to the public is greatly appreciated. I'm not clear if only the validation data is available, or also the data used for the ML fitting, so that should be made clearer. Obviously, it would be appreciated if the whole database would be available to the public.

Response: Thank you. We will, indeed, also provide the whole ML fitting database online upon publication of the work.

Action taken: We updated the Data Availability statement to now read "The potential model described herein as well as the DFT+MBD data used for fitting the model are openly available through the Zenodo repository (DOI: 10.5281/zenodo.4003703)". (Note that this DOI link will be made active upon publication of the work.)

The paper covers an important topic and provides an improved tool to the computational community, to facilitate the investigation of such an elusive, and allotropic-rich, element. This work is well written, clear, and extensively validated. It is worth publishing, if the author can provide a few extra clarifications/details.

Response: Thank you very much.

The ML-part of the work needs more details. While I understand it relies on previous publications, the paper should provide a self-contained description of the ML model used, parameters fitted and specifics on how the reported RMSE were obtained (80-10 split?, 90-10?, cross validation? etc.), even if only in the SI. Also, explicitly telling the number of validation structures versus fitting ones would be interesting.

Response: This is indeed worth adding, and we have done so.

Action taken: In line with the reviewer's comments (and the editorial requests), we moved the details of the methodology to a newly added Methods section at the end of the main text. We also added a footnote to Table 1 giving details of how the RMSE values were obtained and specifying the number of structures in the training and testing sets.

As the forces in Figure 3 are large with respect to what is considered necessary for structural convergence, it is not immediate to translate these results into force-field predictability. It would be more useful to show the same scatterplot type of comparison between the DFT formation energies of the validation structures and the corresponding GAP energies, once the initial structures are allowed to relax using the newly developed force field.

Response: We note that one of the reasons why the forces in Figure 3 are indeed rather large is that these structures are distorted on purpose, to sample more diverse atomic environments. (The energies of the fully relaxed crystal structures are given in Table 2, and

they reproduce the DFT results very accurately, within a few meV per atom.) We now make this clearer in the revised manuscript.

Action taken: To the discussion of Fig. 3, we added a statement that “the test structures are not fully relaxed, on purpose (and neither are those used in the ML fit): they serve to sample slightly distorted environments where there are non-zero forces on atoms” (p. 13).

Lastly, it is common to use force fields to investigate temperature-dependent phenomena. While this work addresses temperature effects in the case of the liquid, it doesn’t discuss how well this new force field performs for solids. For instance, thermal conductivity of black phosphorus nanosheets is of interest. Is this FF a good tool for such a study or not? These types of applications should be discussed.

Response: Thermal conductivity is indeed an important application area of ML force fields, and we envision that the phosphorus model introduced here will be used for that in the future. We are not able to perform a full study of thermal conductivity in the timescale of the present revision, so we would like to avoid undue speculation at this stage. We may, however, mention previous successful studies of the thermal conductivity of silicon with the same ML force field fitting framework. We added a carefully worded discussion to the revised manuscript, as requested.

Action taken: We amended the statement at the end of the corresponding subsection to now read:

“The high accuracy of our ML model for predicting interatomic forces (0.07 eV \AA^{-1} for the 2D configurations; Table 1) allows one to anticipate a good performance for properties that are directly derived from the force constants, *viz.* phonon dispersions and thermal transport, as demonstrated previously for silicon (see refs. 60 and 70, and references therein). A rigorous study of phonons and thermal transport in phosphorene with GAP+R6 is envisioned for the future” (p. 19–20).

Finally, we thank all three reviewers again for their comments, which have helped us to improve the manuscript further.